# Seasonal Variation in Gut Microbiota Related to Diet in *Fejervarya limnocharis*

**DOI:** 10.3390/ani11051393

**Published:** 2021-05-13

**Authors:** Chunhua Huang, Wenbo Liao

**Affiliations:** 1Key Laboratory of Southwest China Wildlife Resources Conservation (Ministry of Education), China West Normal University, Nanchong 637009, China; 18380586112@163.com; 2Institute of Eco-Adaptation in Amphibians and Reptiles, China West Normal University, Nanchong 637009, China; 3Institute of Evolution and Ecology, International Research Centre of Ecology and Environment, College of Life Sciences, Central China Normal University, Wuhan 430079, China

**Keywords:** *Fejervarya limnocharis*, diet, seasons, gut microbiota, function prediction

## Abstract

**Simple Summary:**

Gut microbiota, such as Proteobacteria, Firmicutes, and Bacteroidetes, show chitin-degrading and/or antibiotic biosynthesis ability. To evaluate whether ecological factors (diet) and host conditions (body size, body mass, and body condition) affect the gut microbiota diversity, we analysed the diet composition and host condition in *Fejervarya limnocharis* among different seasons and/or sexes. The dietary difference was not significant among seasons and between males and females, but host condition seasonally changed. Using bioinformatics analysis, we observed that food variations and body mass were significantly correlated with gut microbial composition. Our findings suggest that gut microbiomes of *F. limnocharis* vary seasonally in response to diet variations when facing environmental changes.

**Abstract:**

Organisms adapt to environmental fluctuations by varying their morphology and structural, physiological, and biochemical characteristics. Gut microbiome, varying rapidly in response to environmental shifts, has been proposed as a strategy for adapting to the fluctuating environment (e.g., new dietary niches). Here, we explored the adaptive mechanism of frog intestinal microbes in response to environmental changes. We collected 170 *Fejervarya limnocharis* during different seasons (spring, summer, autumn, and pre-hibernation) to study the compositional and functional divergence of gut microbiota and analysed the effects of seasonal feeding habits and body condition on intestinal microorganisms using 16S rRNA high-throughput sequencing, Tax4Fun function prediction analysis, and bioinformatics analysis. The results showed no significant dietary difference in various seasons and between males and females. However, a significantly positive correlation was detected between dietary diversity and food niche width. Host condition (body size, body mass, and body condition) also revealed seasonal changes. The frogs were colonised by 71 bacterial phyla and dominated by Proteobacteria, Firmicutes, and Bacteroidetes. *Stenotrophomonas* was the most abundant genus in the Proteobacteria. The composition, diversity, and function of intestinal microorganisms in different seasons were significantly different. Significant differences were observed in composition and function but not in the microbial diversity between sexes. Furthermore, seasonal foods and body mass were significantly correlated with gut microbial composition. Our results suggest that gut microbiomes of *F. limnocharis* vary seasonally in response to diet under fluctuating environments.

## 1. Introduction

Animals evolve a variety of behavioral and physiological strategies, including altered feeding and activity patterns and increased mobilization of stored fat to fuel energy demands, to cope with shifting demands [1]. Nutritional demands may vary in response to life-history processes, such as growth, reproduction, and hibernation. The gut microbiome has been proposed as an additional avenue by which animals can cope with changing dietary landscapes and energetic challenges [2,3] and thus play a role that has profound ecological, evolutionary, and environmental implications in amphibians for the most part [4].

Diverse and abundant microbial communities colonise vertebrate digestive systems and coevolve with the host [5,6], which affects multiple aspects of host physiology, including health, nutrition, immunity, development, and reproduction [7,8]. Chitinolytic bacteria (e.g., *Clostridium*) from tadpoles are advantageous in the digestion of insects, which are the main diet of anurans [4]. Important energetic gains made by tadpoles from a fermentative microbial digestion within their guts and these gains, in turn, are expected to be directly proportional to metamorphic size and/or timing [4]. In addition, the intestinal microbiota can develop a natural defense barrier exerting different protective, structural, and metabolic effects on the host epithelium [5]. Particularly, probiotics are proposed as an effective and environmentally friendly alternative to antibiotics, known as beneficial microbes. Antibiotic genes of intestinal microorganisms encode the biosynthesis of antibiotics to affect the intestinal microflora and its antimicrobial and anti-inflammatory effects [9]. However, pathogens carrying antibiotic resistance genes pose a threat to public health [10]. A typical gut microbiome can result in immunological and metabolic disorders [11,12,13]. Consequently, a growing interest focuses on the evaluation of extrinsic and intrinsic factors that manipulate gut microbial communities and diversity [14,15]. Environmental factors are particularly intriguing because they play a role as a major driver of gut microbiota communities [15,16,17,18,19,20,21] and may offer potentially simple mechanisms for treating dysbiosis to improve host health and feed utilisation [16,17].

Host diet is an important environmental factor affecting gut microbial communities [18,19,20,21,22,23,24]. Most previous studies have demonstrated that the gut microbiome seasonally reconfigures to respond to dietary fluctuations [17,24,25,26] and presumably to adapt to new dietary niches. A more diverse diet may improve microbial diversity by providing a more diverse nutrition [27]. Alternatively, diet can indirectly change the microbial composition by varying host physiology, body condition, and immune status [2,28,29,30]. However, whether gut microbiota composition and diversity of amphibians seasonally change in response to diet has not been studied.

Amphibians modify their life-history traits, physiology, behavior, and gut microbiome when they experience variation in environmental factors, such as temperature, rainfall [31,32], and diet [33]. Environmental temperature variations result in changes in food composition of frogs [34,35], thereby affecting their gut microbial composition [36]. *Fejervarya limnocharis*, an amphibian, is widely distributed and can represent a good system for studying the relationship between gut microbiota composition and seasonal variations in terms of adapting to fluctuating diet. Although insects are the major nutrient source of *F**. limnocharis* [33], their gastrointestinal tract is poorly adapted to this food [37]. To compensate, *F**. limnocharis* may depend heavily on their gut microbiota to maximise nutrient extraction from insects [4]. Additionally, given that *F**. limnocharis* are widely distributed in farmlands, their seasonal gut microbiota may be a great system to explore the adaptation of amphibians to human activities and provide advice in the context of conservation biology. In farmlands, compared with frogs in natural environments, *F**. limnocharis* have decreased stomach contents (abundance and number of categories) and increased levels of Firmicutes and Proteobacteria, suggesting that anthropogenic activities have not only affected the food resources of frogs, but also influenced the health and gut microbial ecosystem of wildlife [33].

Here, we investigated the gut microbiota of 161 frogs and stomach contents of 147 individuals, with samples collected over four seasons (e.g., spring, summer, autumn, and pre-hibernation). Using Tax4Fun function prediction, we predicted the function of gut microbiota. We tested for seasonal variation in gut microbiota composition, diversity, and function. We then examined seasonal changes in diet composition and diversity in the host under the four seasons. Finally, we analyzed the effects of ecological factors (e.g., diet) and host conditions on the microbiota diversity of frogs to understand how seasonal factors may drive fluctuation in the microbiome composition and function.

## 2. Materials and Methods

### 2.1. Sample Collection

A total of 170 adult frogs were collected during spring, summer, autumn, and pre-hibernation (118 males and 52 females) in Xishan from September 2018 to November 2019 in Nanchong city (106°2′ E, 30°47′ N; 310 m altitude), Sichuan Province, China (Appendix A). All the frogs were collected in a cool box with two ice bags and immediately transported to the laboratory of China West Normal University for immediate experiments. The gender of all individuals was confirmed by direct observation of secondary sexual characteristics [38]. We sacrificed all individuals using single pithing. Body size (snout–vent length (SVL)) was measured to the nearest 0.01 mm with calipers, whereas body mass was measured to the nearest 0.1 mg with an electronic balance. We calculated the body condition of each individual based on the ratio between log of body mass and log of body length. The gut content was collected within 20 min after pithing. We carefully isolated the digestive tract from the body and then collected the gut and stomach. The gut contents within each individual were emptied into a sterile vial and immediately stored at −80 °C. The specimens were preserved in 4% buffered formalin, and the stomach was stored in 70% ethanol.

### 2.2. Diet Analyses

The prey items of 147 stomachs were identified at the order level under a stereomicroscope. Then, we estimated the percentage of frequency of each prey category. For the assessment of diet diversity, the Shannon–Weiner index was calculated based on the following model:(1)H=−∑(Pi) Ln(Pi),
where *H* represents the Shannon–Weiner index; the *P*i is the proportion of individuals found in the *i*th species. In a sample the value of *P*i is unknown but is estimated as n*_i_*/N, in which *N* is the total number of all species, and Ni is the number of *i*th species. [39]. In addition, the Simpson diversity index was used to estimate the niche width of stomach contents [39,40], and the formula is as follows:(2)B=1∑Pi2,
where *B* expresses the Simpson’s index; the *P*i is the proportion of individuals in the *i*th species.

The numbers, frequency of each prey category, food diversity and ecological niche of stomach contents between sexes among different seasons were compared using non-parametric statistical methods, Pearson’s correlation analysis, and Wilcoxon rank-sum test. All statistical analyses were conducted using R 3.6.2.

### 2.3. DNA Extraction and Sequencing

DNA extraction and sequencing were operated by Novogene Corporation (Novogene Corporation, Beijing, China). We extracted DNA from entire intestines from 161 *F. limnocharis* using TIANgen DNA extraction kit DP328 (Tiangen, Beijing, China). We amplified the V4 region of 16S rRNA gene with a specific primer set 515F (5′-GTGCCAGCMGCCGCGGTAA-3′) and 806R (5′-GGACTACHVGGGTWTCTA AT-3′). All polymerase chain reaction (PCR) reactions were performed in bio-Rad T100 gradient PCR instrument with 15 μL Phusion^®^ High-Fidelity PCR Master Mix (New England Biolabs), 0.2 µM forward and reverse primers and 10 ng template DNA per 30 μL reaction. The PCR cycle consisted of initial denaturation at 98 °C for 1 min; 30 cycles of denaturation at 98 °C for 10 s, annealing at 50 °C for 30 s, extension at 72 °C for 30 s; and extension at 72 °C for 5 min. The PCR products were mixed with same volume of 1 × loading buffer (contained SYB green) and were subsequently subjected to electrophoresis on 2% agarose gel for detection. The PCR products were mixed in equidensity ratios. Then, we obtained the purified PCR products using GeneJETTM Gel Extraction Kit (Thermo Scientific, Shanghai, China).

Sequencing libraries were generated using Ion Plus Fragment Library Kit (48 rxns) (Thermo Scientific, Shanghai, China). The library quality was assessed on the Qubit@ 2.0 Fluorometer (Thermo Scientific, Shanghai, China). Finally, the library was sequenced on an Ion S5TM XL platform and 400 bp single-end reads were generated.

### 2.4. Operational Taxonomic Unit (OTU) Cluster and Species Annotation

We assigned single-end reads based on their unique barcode and truncated them by cutting off the barcode and primer sequence to the samples. Quality filtering of raw reads was performed under specific filtering conditions in accordance with the Cutadapt quality-controlled process ([41]; V1.9.1, http://cutadapt.readthedocs.io/en/stable/; access date: late January to early February 2020). To detect chimera sequences, we used UCHIME algorithm (UCHIME Algorithm, http://www.drive5.com/usearch/manual/uchime_algo.html/; access date: late January to early February 2020) to compare the reads with the reference database (Silva database, https://www.arb-silva.de/; access date: late January to early February 2020) and removed the chimera sequences. Finally, we obtained the clean and effective reads using subsequent analysis.

We clustered all the effective sample data using Uparse software (v7.0.1001) and assigned the sequences with ≥97% identity to the same OTU. The representative sequence for each OTU was screened for further annotation. For each representative sequence, we used the Silva Database (https://www.arb-silva.de/; access date: late January to early February 2020) to annotate taxonomic information based on Mothur algorithm. We also used annotation analysis (threshold value: 0.8–1) to obtain taxonomic information and count the community composition of samples at all levels, such as kingdom, phylum, class, order, family, genus, and species, based on the Mothur algorithm and SSUrRNA database of SILVA132 (https://www.arb-silva.de/; access date: late January to early February 2020). OTU abundance information was normalized using a standard sequence number corresponding to the sample with the least sequences. Finally, we performed subsequent analysis of alpha and beta diversities based on this output normalized data.

### 2.5. Data Analysis

Alpha diversity indexes, which is used to analyse the complexity of bacterial diversity for a sample, was calculated with QIIME (Version 1.9.1) and displayed with R software (Version 2.15.3), such as observed-species index, Chao1 index, Shannon-Wiener index, Simpson’s index, ACE (Abundance Coverage-based Estimator) index, and good-coverage index (http://www.mothur.org/wiki/; access date: late January to early February 2020). To evaluate the differences among samples in terms of bacterial community complexity, we calculated the beta diversity on weighted and unweighted UniFrac using QIIME (Version 1.9.1). Moreover, to find differences among different samples (groups), we used the Unweighted Pair-group Method with Arithmetic Mean (UPGMA) and analysis of intra (inter)-group differences in beta diversity index. In particular, the NMDS (nonmetric multidimensional scaling) ordination approach was performed to visualize beta-diversity patterns and reflect the inter- and intra-group differences based on Bray–Curtis dissimilarity distances. UPGMA clustering was performed as a type of hierarchical clustering method to study the similarities between different samples using average linkage and conducted by QIIME software (Version 1.9.1).

The between- and within- group difference based on alpha and beta diversity indexes and bacterial abundance were analysed by Anosim, (un)weighted UniFrac Wilcoxon rank-sum test and linear discriminant analysis effect size (LEfSe) analysis, whereas the different bacteria was found by parametric test (*t*-test). Through sample clustering tree display, the differences in the community structure between different samples or groups were explored.

Data on food frequency and microbiota β-diversity were analyzed using the Procrustes routine in QIIME (Version 1.9.1). On the basis of the frequency of prey item and microbiome diversity data, weighted and un-weighted UniFrac PCoA analysis were combined with Procrustes analysis to visualize the relationship between diet and intestinal flora, and evaluated whether diet affects the gut microbiota abundance. Relationship significance was tested with Monte Carlo test in QIIME (Version 1.9.1). Generalized Linear Models (GLM) analysis with SPSS was also used to test the dependence of bacterial diversity and determine the effect of diet diversity, body size, body mass, and/or body condition on the microbiota.

### 2.6. Function Prediction

The function of intestinal microbiomes was predicted using Tax4Fun functional prediction analysis based on the minimum 16S rRNA sequence similarity by extracting the KEGG database prokaryotic whole genome 16S rRNA gene sequence. The predicted function was aligned to the SILVA SSU Ref NR database using BLASTN algorithm (BLAST Bitscore >1500) to establish a correction matrix. Finally, the prokaryotic whole genome functional information of the KEGG database annotated by UProC and PAUDA was mapped to the SILVA database to implement the SILVA database function annotation. The sequenced samples were clustered out of the OTU using the SILVA database sequence as a reference sequence to obtain functional annotation information.

## 3. Results

### 3.1. Gut Microbiota Composition of F. limnocharis

The V4 region of bacterial 16S rRNA gene in intestinal microbiota collected from 161 frogs across four seasons was sequenced to characterize microbiotas of *F. limnocharis*. All sequences could be identified to two boundaries, 71 phyla, 83 classes, 187 orders, 381 families, and 1301 genera. At the phylum level (Appendix A, Figure 1b), Proteobacteria, Firmicutes, and Bacteroidetes had the highest abundance and contributed 43.58%, 41.11%, and 5.78% of bacteria, respectively. At the genus level (Appendix A, Figure 1e), *Stenotrophomonas* (phylum: Proteobacteria), *Faecalitalea* (phylum: Firmicutes) and *Parabacteroides* (phylum: Bacteroidetes) were the most abundant and accounted for 36.69%, 11.09%, and 5.78% of bacteria, respectively.

### 3.2. Seasonal Changes of Intestinal Microbiota in Frogs

The seasonal variation in the abundance and diversity of intestinal microbiota was significant (ANOSIM: *r* > 0, *p* < 0.001). The unique OTUs in summer were the most abundant (Appendix A). The alpha diversity metrics (observed taxa and Chao 1) revealed the highest diversity and richness of the intestinal microbiota in summer and the least in autumn (Figure 2b,d; Appendix A, for detailed information on the alpha diversity of gut in all samples, see Appendix A). Shannon diversity index and Simpson’s diversity index were the lowest in spring (Figure 2a,c). Inter- and intra-species differences in the microbiota among seasons were also explored (Anosim: *r* > 0, *p* < 0.01; Appendix A). LEfSe analysis revealed that g_*Stenotrophomonas* was the top genus-level bacteria distinguished in frogs during different seasons (Appendix A).

The seasonal relative abundance and distribution of bacteria may be indicative of seasonal variation in microbiota composition. Particularly, the composition of gut microbiota was seasonally changed (Figure 1a–f and Figure 3a,b). Differences were found in the specific bacteria among individuals (Appendix A). The significant phylum and genera (*p* < 0.05) are listed in Appendix A, respectively. The abundance of chitin-digesting bacteria was found to be high in spring (Figure 3c).

After intrinsic differences in the gut microbiota composition were established between females and males within each season, significant differences were observed in Enterobacteriales (*p* = 0.032) order, Enterobacteriaceae (*p* = 0.032), and Peptococcaceae (*p* = 0.014), unidentified Coriobacteriales (*p* = 0.040), Phaselicystidaceae (*p* = 0.016), Aerococcaceae (*p* = 0.044) family, *Raoultibacter* (*p* = 0.040), *Phaselicystis* (*p* = 0.016), *Variovorax* (*p* = 0.034) genus, and *Variovorax paradoxus* species (*p* = 0.034) in autumn.

### 3.3. Variations of Function Prediction in Different Seasons

Functional predictions via Tax4Fun have suggested that the following three predicted functions dominated the four seasons: metabolism, genetic information processing, and environmental information processing (Appendix A). However, we found that most predicted functional categories in KEGG (Kyoto Encyclopedia of Genes and Genomes) pathways were significantly different among the four seasons, such as metabolism, environmental adaptation, and immune system (Figure 4a; *t*-test, *p* < 0.05; see Appendix A). Particularly, metabolism (e.g., amino acid and lipid metabolism) and diseases (e.g., infectious diseases) were significantly higher in spring and summer than in autumn and pre-hibernation (Figure 4b; Appendix A).

In order to further understand bacterial contribution to the specific environmental adaptability and diseases, we predicted that the gene function was linked to chitin-degrading ability, the biosynthesis of antibiotics, and antibiotic resistance among the four seasons. Analysis on potential chitin-degrading ability showed that the abundance of chitin-degrading enzymes, such as chitinase, β-N-acetylglucosaminidase, α-N-acetylglucosaminidase, chitin deacetylase, or putative chitinase, was significantly different among seasons (Kruskal-Wail test, *p* < 0.01) (Figure 5). Antibiotic genes were also significantly different between spring and summer (Mann-Whitney U test, *p* < 0.001), between spring and autumn (Mann-Whitney U test, *p* < 0.001), between spring and pre-hibernation (Mann-Whitney U test, *p* < 0.001), and between summer and pre-hibernation (Mann-Whitney U test, *p* = 0.001). In particular, the total abundance of all antibiotic genes was enriched in pre-hibernation as compared to other seasons (Wilcoxon rank-sum test, *p* < 0.01; Appendix A). Additionally, significant differences were also found in antibiotic resistance gene (beta-Lactam_resistance) between summer and autumn (Mann-Whitney U test, *p* < 0.001) and between summer and pre-hibernation (Mann-Whitney U test, *p* < 0.05).

The potential function of the intestinal microbiota was compared between females and males in autumn. In female frogs, nucleotide metabolism was significantly higher in autumn (*t*-test, *p* = 0.038). The total abundance of chitinase and antibiotic genes in females was higher than that in males (Appendix A). Chitin-degrading enzyme (putative chitinase and bifunction chitinase/lysozyme) and antibiotic genes tetracycline biosynthesis and clavulanic acid biosynthesis) were significant different between females and males in autumn (Wilcoxon rank-sum test: putative chitinase, *p* = 0.036; bifunction chitinase/lysozyme, *p* = 0.016; tetracycline biosynthesis, *p* = 0.040; and clavulanic acid biosynthesis, *p* = 0.041). In addition, the vancomycin resistance gene was also a significant difference between females and males in autumn (Wilcoxon rank-sum test, *p* = 0.043).

### 3.4. Dietary Changes

Prey items in stomachs were sampled in 147 frogs (successfully identified for 114 frog specimens) to construct a concordance relationship between diet composition and gut microbiota. A total of 373 prey items were identified to six classes and 18 orders (Appendix A). After the variability within and between seasons was established (for detailed information of stomach contents during four seasons, see Appendix A), the seasonal and sexual variations in food niche width and diet diversity were analysed. A positive correlation was found between seasons (Pearson: *r* = 0.97; Wilcoxon rank-sum test: *p* < 0.05). Among the seasons, the largest food niche width and diet diversity was found in autumn (Appendix A). Furthermore, food niche width was positively related to the food diversity of females and males in summer (Pearson: *r* = 0.76; Wilcoxon rank-sum test: *p* < 0.05), and food niche width in males was larger than that in females (Appendix A). However, no significant differences in food composition, food diversity, and niche width were detected among seasons and between males and females in summer (Anosim analysis: *r* > 0; Wilcoxon rank-sum test: *p* > 0.05).

### 3.5. Explanatory Factors of Intestinal Microbiota

Bioenv analysis based on beta diversity in bacteria was used to identify the optimal combination of ecological factors (diet diversity, humidity, and pH) and explain the seasonal variation of gut microbiota. Diet was determined to be the optimal combination (Bioenv: size = 1, correction = 0.6571).

Procrustes analysis of food frequency and microbiota β-diversity was conducted on 113 individuals during various seasons to co-visualise the data (Figure 6; Appendix A). Separations based on either diet or microbiota co-segregated along the first axis of both data sets (weighted UniFrac, Figure 6a; unweighted UniFrac, Figure 6b). A strong relationship was observed between diet and intestinal microbiota (the Monte Carlo test; weighted: Counter_better = 19, M^2^ = 0.957, *p* = 0.019; unweighted: Counter_better = 2, M^2^ = 0.945, *p* = 0.020).

In addition to ecological factor (diet), the effect of body size (SVL), body mass, and body condition on gut microbiota was also explored. Body size, body condition, and body mass (converted by log 10 for the normal distribution) significantly shifted between seasons (*t*-test: *p* < 0.01, except between spring and autumn). GLM analysis was used to evaluate whether diet and body mass affects the microbiota and found a significant negative association (Pearson: *r* < 0; GLM: *p* < 0.01; Table 1)

## 4. Discussion

Body size, body condition, and diet are associated with gut microbiota [33,42]. Consistent with previous studies, our study found that diet (frequency of prey item and dietary diversity) and body mass were substantially associated with gut microbiota diversity. Interestingly, we rarely found an impact of temperature and humidity on gut microbial variation. These seasonal changes in bacteria could be explained by variations in seasonal food availability.

Consistent with other frogs (*F. limnocharis* [33], *Babina adenopleura* [43], *Polypedates megacephalus* [25], *Odorrana tormota* [44], and *Rana dybowskii* [19,45], Proteobacteria, Firmicutes, and Bacteroidetes detected in the intestinal flora were considered to be the main bacteria. This implies that the endogenous environment selects microbes that are optimally fit for the intestinal characteristics. On the other hand, the variation of community diversity found in *F. limnocharis* was similar with that found in *R. dybowski*i [46], but different from *P. megacephalus* [25] and *O. tormota* [44]. In our findings, frog communities were more diverse and abundance in summer compared to other seasons. This pattern suggested that the environmental conditions played major roles in determining the gut microbial composition.

Diet is the main ecological factor influencing bacterial community changes in frogs [33,47]. Changes in dietary habits can seriously affect the structure and composition of intestinal microbial flora [13]. Our founding revealed that the first PCA axis (PC1) explained 19.94% of the relationship between food frequency and microbiome diversity. Particularly, frogs with less digestible diet (low level of nutrient) had long digestive tract length [35,48]; this practice may indirectly affect gut microbiota composition as gut microbiota diversity was associated with intestinal pattern [27,36]. *Stenotrophomonas* (Gammaproteobacteria) was linked to chitin digestion [44] and is the main bacterium in *F. limnocharis*. In spring, when frogs eat more Scolopendromorpha and Coleoptera, the abundance of chitin-digesting bacteria increases. These results suggest that chitinolytic bacteria may confer digestive advantages to frogs. Moreover, we found more Lachnospiraceae (Firmicutes) linked to lipid metabolism in autumn and pre-hibernation as compared to that in both spring and autumn, which was likely to relate to the energy reserve (e.g., fat body formation) in preparation for hibernation [13]. The variations in dominant gut flora taxa at different seasons implied that *F. limnocharis* frogs acquire different bacteria due to variations in their seasonal diet. Nevertheless, intestinal microorganisms do not affect the dietary preference of frogs [49] and instead allows them adapt to their dietary niche [39].

In addition to metabolic adaption to diet, microbiome found in this frog was likely to reflect the status of the health and diseases of the host species among seasons. Pathogens (e.g., *Pseudomonas* and *Batrachochytrium dendrobatidis*) result in illness or even death in frogs [45,50,51] whereas probiotic bacteria (e.g., *Bifidobacteria longun* and *Lactobacillus casei*) and bacteria associated with pathogen resistance may promote host health [50,52]. More Enterobacteriaceae, as a common marker of gut dysbiosis, was found in frogs in spring compared to other seasons, suggesting that frogs have lower of SCFAs and higher acidic environment and diseases in spring [53]. Low gut microbiota diversity in spring supported this assumption because low gut microbiome diversity exhibited increased stress responses (high glucocorticoid levels) and reduced immune function (few cells that secrete local, strain-specific immunoglobulin A) [28,54,55].

At the function level, our study showed that many bacterial genes involved in amino acid metabolism, lipid metabolism, and chitin digestion were found during spring and summer and when the host had a low immune system, suggesting the occurrence of an energy tradeoff. By contrast, many antibiotic genes linked to antibiotic biosynthesis and gene involved in carbohydrate metabolism and cell motility was observed during autumn and pre-hibernation. This stimulation of bacterial energy metabolism and cellular activity could reflect the high SCFAs productions by gut bacteria [56] to provide frogs with additional energy in periods of pre-hibernation when nutrients are restricted. For example, in *Rana dybowskii*, the Firmicutes/Bacteroidetes ratio decreased from 4.23 in summer to 0.63 in winter to adapt to the challenging environment [46].

Furthermore, frog microbiota is composed additionally of functions which are related to antibiotic resistance [10]. Consistent with a previous study, we identified resistance gene (vancomycin resistance gene and beta-Lactam_resistance gene) in *F. limnocharis*, suggesting that wild frogs are reservoirs of multidrug-resistant bacteria [10]. The accumulation of antibiotic resistance genes in intestinal environment may increase the risk of pathogen and pose a threat to public (human) health, especially vancomycin-resistant gene [9,10]. Therefore, frogs were regarded as bioindicators when they reflected the health of their environmental changes. The emergence of antibiotic resistance genes in frogs may be resulted from the absorption of drug residues by the skin, the ingestion of invertebrates that consume drug residues, and/or the entry into the frog’s food chain of sewage that calls for antibiotics [10].

In summary, the gut microbiome of *F. limnocharis* exhibits seasonal variation and can respond rapidly to changes in host diet. This study suggests that the gut microbiota is an important system that provides dietary and metabolic flexibility for the host and might influence host acclimatization to fluctuating environments [57,58,59]. In addition to the profound ecological and evolutionary implications, the gut microbiota are regarded as a system contributing to host phenotypes [4,42,60] and evolution [5]. Changes in the gut microbiota community may be an important adaptive mechanism to some extent.

## 5. Conclusions

The gut microbiomes of *F. limnocharis* exhibit seasonally variations in response to changes in its diet (frequency and diversity) and body mass under fluctuating environments. These shifts may help *F. limnocharis* cope with seasonal changes in food available and maintain energy balance while living in a challenging environment. The findings suggest that gut microbiome is an important system that promotes metabolic flexibility and adaption to fluctuating environments. Further research on *F. limnocharis* could explore the influence of food nutrients/chitin content on intestinal microorganisms to uncover how the microbiota allows these frogs to occupy special dietary niches.

## Figures and Tables

**Figure 1 animals-11-01393-f001:**
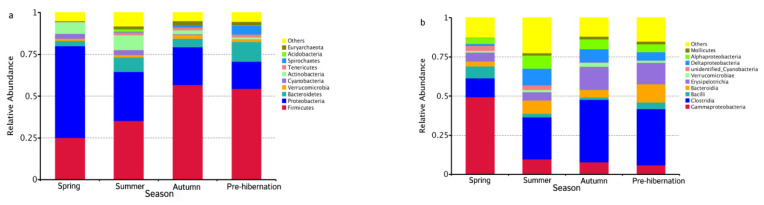
The relative abundance of bacterial compositions in different seasonal groups at different levels, phylum (**a**), class (**b**), order (**c**), family (**d**), genus (**e**), and species (**f**).

**Figure 2 animals-11-01393-f002:**
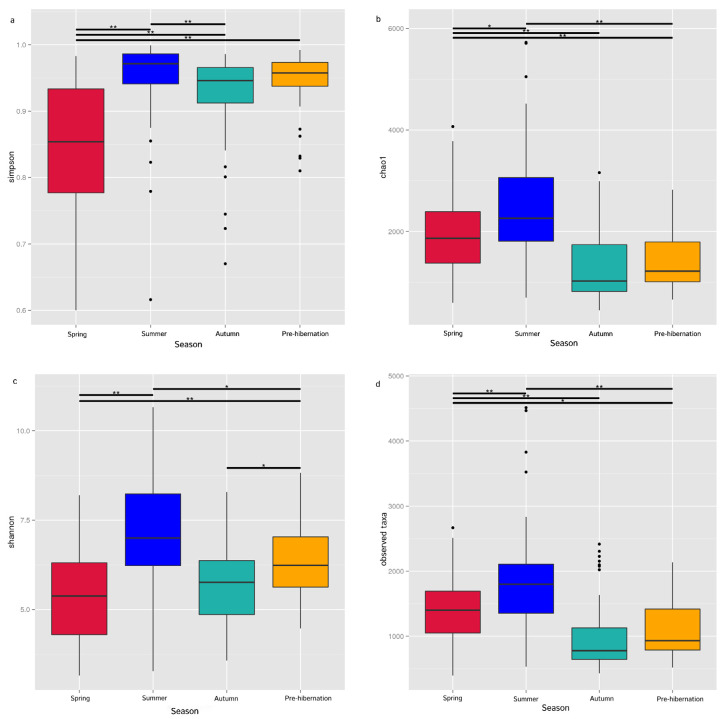
Comparison of alpha diversity of the gut microbiota in *F. limnocharis* at different seasons (Wilcoxon rank-sum test; *p* < 0.01 **; and *p* < 0.05 *). Simpson (**a**), Chao1 (**b**), Shannon (**c**) and observed taxa (**d**).

**Figure 3 animals-11-01393-f003:**
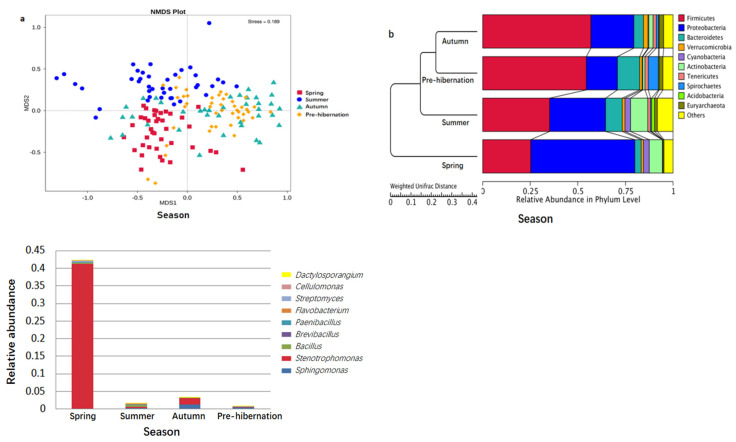
Comparation of gut microbiotal composition across four seasons. (**a**) NMDS analysis between different individuals based on OTUs. Each point in the graph represents a sample, the distance between points indicates the degree of difference, samples in the same group are represented in the same color, when the stress <0.2, NMDS can accurately reflect the differences between groups and within groups of samples. (**b**) The UPGMA cluster analysis of different seasonal samples in weighted UniFrac distances. The UPGMA cluster tree structure is shown on the left side of the figure; the relative abundance distribution of each sample at the phylum level is shown on the right side. Others represent the sum of the relative abundances of all the other phyla outside the 10 phyla in the graph. (**c**) Comparison of chitin-digesting bacteria in *F. limnocharis* at different seasons.

**Figure 4 animals-11-01393-f004:**
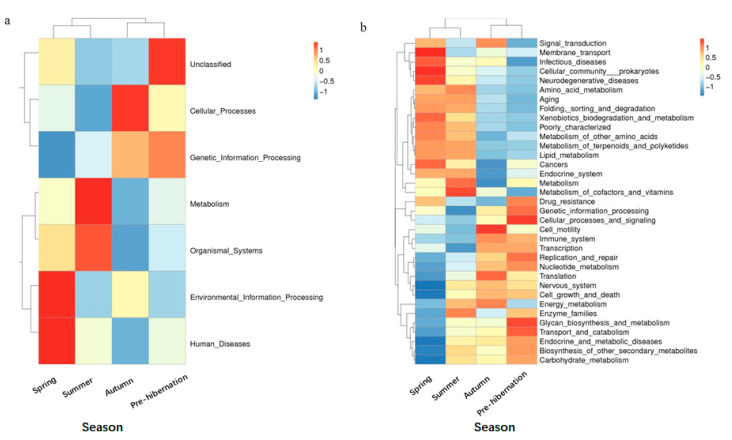
Cluster heatmap of annotated function by Tax4Fun. (**a**) The main function in level 1. (**b**) The metabolism functions in level 2. In the figure, the horizontal ordinate represents the sample information and species annotation information; the cluster trees on the left and the top are species clustering and sample clustering, respectively; middle heat map matching is the Z-value. Z-value is obtained after standardized treatment of relative abundance of species. It is the difference between the relative abundance of a sample in this classification and the average relative abundance of all samples in this classification divided by the standard deviation of all samples in this classification.

**Figure 5 animals-11-01393-f005:**
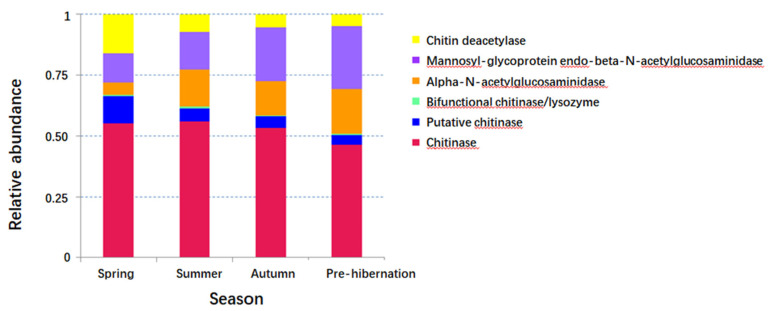
Abundance of Tax4Fun-predicted reads annotated to genes for chitin-degrading enzyme.

**Figure 6 animals-11-01393-f006:**
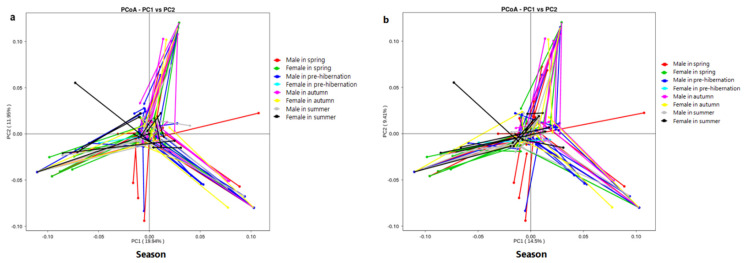
The relationship between food types and gut microbes. (**a**) Procrustes analysis combining weighted UniFrac PCoA; (**b**) Procrustes analysis combining unweighted UniFrac PCoA. The abscissa is the first principal component and the percentage represents the contribution of the first principal component to the sample difference; The ordinate is the second principal component and the percentage represents the contribution of the second principal component to the sample difference; Each point in the figure represents an individual and samples in the same group are represented by the same color.

**Table 1 animals-11-01393-t001:** Results of a GLM (Generalized Linear Models) analysis testing whether microbial diversity depends on diet and body mass in *F. limnocharis.*

Coefficients	Wald Chisq	df	*p*
Intercept	1306.936	1	<0.0001
Diet diversity	14.279	2	0.001
Body mass	1597.758	88	0.0001

df is the degree of freedom; and *p* is the probability value, which is used to show the level of significance. When *p* < 0.05, it means there is a significant difference.

## Data Availability

The data presented in this study are available in the Appendix A.

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
