# Peer review of "Seasonal Variation in Gut Microbiota Related to Diet in *Fejervarya limnocharis"

_animals, 2021, doi:10.3390/ani11051393_

Round 1

Reviewer 1 Report

This research is of potential interest to a range of audiences but the manuscript needs to be overhauled entirely before it can be published.

Throughout the entire manuscript the English is poorly written which makes it difficult to comprehend some of the salient points that the authors are trying to convey. The manuscript needs to be completely rewritten with the use of a native English speaker before it is at a level that will be acceptable for publication. I also have concerns regarding how the frogs were treated prior to sampling, I’m not sure if this is a true issue or a result of problems associated with language difficulties but this may render the research unethical and thus unsuitable for publication. The statistical information and tests are severely lacking and need significant bolstering so that the reader can fully understand the results.

Lines 100-113: Although this information is interesting from an ecological perspective, I’m not sure why you have included it as you are not testing for the effects of locality on your frogs, all animals were captured from a single site. This would be more relevant if you were looking at location as a factor of microbiota composition.

Line 123: Frogs were placed into dry ice while they were alive? If that is the case, then that is not ethical and therefore this research cannot be accepted for publication.

Lines 142-144: Did you test for normality before choosing these tests?

Lines 197-200: Again, did you test for normality? What do you mean you explored Beta diversity, you need to be specific about the tests that were used. Do you mean to say you used t-tests in your PCoA analysis – this is confusing as written and would not be a standard way of analysing a PCoA, most commonly it would be using PERMANOVA or Adonis testing.

Figure 2: I find the figures a little confusing. On the X-axis why don’t you simply have the label as “Season” and then label the groups accordingly, i.e. summer, spring etc., this would make it much easier to read.

Figure 3: What do each of the graphs represent, you have not told us what (a), (b) or (c) are, I know that you have it on the graph, but it must also be included in the figure caption. Also please replace the letters on the X-axis with the name of the season.

Lines 289-294: Are you just looking at the graphs and saying that there is greater diversity or have you shown this statistically? We don’t know as you have not included any of this information in the document.

Lines 295-301: What do you mean by smaller? I can’t see anywhere that you have done any Beta dispersion analysis so without this information you can’t make this statement. Also when you are comparing multiple groups such as this you need to do pairwise analysis to make sure that there is difference/similarity between all combinations of groups.

Author Response

Thanks for you take the time to carefully review the article. Your suggestions are very valuable. We responded them in " Responses of comments of reviewer1".

Reviewer 2 Report

Dear authors,

as I have comented intensively and in detail the attached pdf-file, I refrain here to the main issue.

I recognize that the authors have done a significant amount of data collection and analyses, all well designed and done. However, when reading the introduction (which is far too long and without a clear focus) I wondered, why it is interesting to perform this study, i.e., which scientific problem is expected to be solved. I strongly recommend to rewrite and shorten the whole introduction to raise more interest on what is following.

M&M is covering nearly everything needed.

Results are presented (too) exhaustive. I do not like to place many results to the supplementary material - if something is important for understanding, it should be in the main text, if not, you should consider to delete it completely.

Discussion: a lot of repetition of the results which should be avoided. The whole discussion would benefit from a bit more structure, less repetition, focus on amphibians and being more straightforward.

Conclusion: It is not surprising that the gut microbiome varies with diet and season. If not, it would have been very strange. So you have to work out better, what is the new contribution of this paper to science making it worth to publish.

Please, rethink the presentation and rewrite the ms in a considerably more focussed way to give it more appeal for a broader audience.

Author Response

Thanks for you take the time to carefully review the article. Your suggestions are very valuable. We have sorted out your questions and responded them in "Responses of comments of reviewer 2".

Round 2

Reviewer 1 Report

While the manuscript is much improved in terms of language and clarity, some work still needs to be done on the English in order to make it suitable for publication. There are still numerous grammatical and phrasing issues that occasionally make it difficult to read. I have not highlighted these in this review as that is best left to copy editing.

Abstract

Lines 22-24: I’m not sure what the authors are trying to say with this sentence

Introduction

Line 62: Do you mean environment treats dysbiosis? This is very awkward as written

Line 65: As written it implies this is the case in ALL animals, but we don’t have enough data to support such a statement.

Lines 68-71: Remove this statement, it does nothing to add to the introduction

Line 72-73: Remove “thus changing microbial diversity” and line about fish

Line 89: What is meant by stomach contents? This is ambiguous, do you mean they eat less, have a less diverse range of prey items, etc.

Line 100-103: Remove this final statement, you don’t need to summarise your findings here. The final sentence in this paragraph should be to outline the importance of your study.

Line 109: I’m still unclear of the shipping methods. Do the authors mean that the frogs were transported in cool boxes with ice or that they were placed into a bag of ice?

Results:

Line 220: Change richness to abundance

Lines 225-232: Just state your results, the links to other disease or functionality should be raised in the discussion section.

Figure 2:

Here you have shown an NMDS plot but I can’t see any mention of this in your methods. You stated that you did a PCoA plot which you have put into the supp material? Also you should remove the elipses it is not recommended to include these as they can wrongly influence how readers interpret data.

The analysis and rationale for section 3.3 is really unclear as well as some of the systems being investigated such as “human diseases”. This needs to be clarified or removed.

Lines 291-293: Antibiotic genes? This is the first mention of this in the manuscript. You need to include this into your introduction and methods if you are planning on looking into it. We will need some justification as to why you are doing this. Without this information there is no context and it seems random and out of place.

Discussion

Line 352: PCA? All I have seen is a figure for an NMDS but you mentioned in your methods a PCoA which has been hidden in the supp material? You need to be consistent and accurate with the tests you have used. At the moment you have mentioned 3 different statistical tests. If you are referring to the PCoA plot to explain your results it warrants being included in the main text.

Line 363-364: You have no evidence that the frogs acquire these bacteria, they may already be present but preferentially selected for within the existing microbiome due to an increase in dietary precursors.

In general the discussion is too short. We need further comparisions of this study to other amphibian studies to highlight differences and similarities. We also need you to expand on some of the physiological and ecological implications of your findings.

Author Response

Dear professor,

         Thanks for taking the time to read our manuscript and for your valuable comments. These comments are very important to us. We tried to solve these problems. 

         Please see attached for more details.

Best regards

Professor Wen Bo Liao

Reviewer 2 Report

The ms is considerably improved and has gained structure. Thank you for considering widely my recommendation. Final comment: the correct name of your alpha diversity measure is: Shannon-Wiener-Index

Author Response

Dear professor,

       Thanks for your comment, we have corrected the name of Shannon-Wiener-Index. Details are as follows:

Point 1: Final comment: the correct name of your alpha diversity measure is: Shannon-Wiener-Index.

Response 1:

In line181-183: we change “observed-species, Chao1, Shannon, Simpson, ACE and good-coverage” to “observed-species index, Chao1 index, Shannon-Wiener index, Simpson’s index, ACE index and good-coverage index” (Page: 4, line: 186-188).

In line 237: we change “Shannon and Simpson indices” to “Shannon diversity index and Simpson’s diversity index” (Page: 6, line: 254).

Best regards

Professor Wen Bo Liao